# Systemic Inflammatory Markers Are Predictive of the Response to Brachytherapy in the Prostate

**DOI:** 10.3390/cells9102153

**Published:** 2020-09-23

**Authors:** Daniel Taussky, Denis Soulieres, Miguel Chagnon, Guila Delouya, Houda Bahig

**Affiliations:** 1Department of Radiation Oncology, Centre hospitalier de l’Université de Montréal (CHUM), Montreal, QC H2X 0C1, Canada; guila.delouya.chum@ssss.gouv.qc.ca (G.D.); houda.bahig.chum@ssss.gouv.qc.ca (H.B.); 2Département Hématologie-oncologie, Centre hospitalier de l’Université de Montréal, Montreal, QC H2X 0C1, Canada; denis.soulieres.chum@ssss.gouv.qc.ca; 3Département de Mathématiques et de Statistique, Université de Montréal, Montreal, QC H2X 0C1, Canada; chagnon@DMS.UMontreal.CA

**Keywords:** prostate cancer, inflammation, neutrophil to lymphocyte ratio, platelet to lymphocyte ratio

## Abstract

We analyzed the influence of the neutrophil/lymphocyte ratio (NLR) and platelet/lymphocyte ratio (PLR) on the biochemical recurrence (BCR) in low-intermediate risk prostate cancer (PCa). A total of 604 patients treated with exclusive brachytherapy for low- and intermediate-risk cancers were included in this study. No patient received either androgen deprivation or brachytherapy as a boost. BCR was defined according to the Phoenix definition (nadir prostatic specific antigen (PSA) +2). The median follow-up was 60 months (IQR 44–48 months). An NLR > 3 was more frequent in statin users (*p* = 0.025), but not in diabetics (*p* = 0.079). In univariate analysis (UVA) and multivariate analysis (MVA), a NLR > 3 (MVA *p* = 0.03), as well as Cancer of the Prostate Risk Assessment (CAPRA) low- vs. intermediate-risk (MVA *p* = 0.04), were predictive of BCR. When combining the NLR score with the CAPRA risk group, CAPRA intermediate risk patients with an NLR ≤ 3 (*n* = 157) had the worst (*p* = 0.0276) BCR rates, with a 5-year recurrence-free survival (*p* = 0.004, Bonferroni correction for six comparisons *p* = 0.024). We were able to identify a subgroup of PCa patients with CAPRA intermediate-risk and an NLR ≤ 3 who had worse BCR. This is in contrast to most other cancers, which have a worse prognosis when the NLR is high.

## 1. Introduction

The management of low- and lower tier intermediate-risk prostate cancer (PCa) is controversial because of the heterogeneous nature of these cancers. The best radio-oncological treatment for these patients has not been established. Treatments vary, between external radiotherapy with or without androgen deprivation therapy (ADT) and brachytherapy as an exclusive treatment or as a boost to external radiotherapy, depending on the presence of several risk factors.

Several studies have suggested an association between chronic inflammation and PCa [1]. Several cytokines are involved in prostate inflammation, most notably interleukin (IL)-6, which has been implicated in regulating the progression of PCa [2]. Markers of systemic inflammation appear to generally have a significant prognostic value, but results are somewhat contradictory. Some studies showed that inflammation is associated with more aggressive disease, with a higher risk of metastasis and a higher rate of biochemical progression [3]. On the other hand, local inflammation in the prostate seems to have a protective effect [4,5].

The neutrophil to lymphocyte ratio (NLR) is an independent predictor of recurrence in several cancers [6]. The importance of NLR as a prognostic factor is well-established for PCa. In a meta-analysis by Tang et al. [7] of PCa patients, a higher NLR was found to be associated with a lower overall survival (OS) and recurrence-free survival. Another marker of systemic inflammation is the platelet to lymphocyte ratio (PLR). Similarly, a high PLR has been shown to be associated with worse OS in various solid cancers in a systematic review of databases [8].

The aim of this study was to assess the influence of the blood inflammatory markers in the complete blood count (CBC) on recurrence-free survival in low-intermediate-risk PCa. In doing so, we aimed to identify a subgroup of patients that could benefit from a dose escalation or additional treatment, such as hormonal therapy or a combination of external beam radiotherapy.

## 2. Material and Methods

### Study Population

Research and ethics board approval (REF 19.235) by the Centre Hospitalier de l’Université de Montréal was obtained for this study on December 2nd, 2019. From our institutional database, 628 patients who had undergone permanent seed brachytherapy alone at the Centre Hospitalier de l’Université de Montréal (CHUM) from April 2005 and August 2019 were identified.

Two physicians performed all brachytherapy implants with Iodine-125 seeds via 3-D trans-rectal ultrasound-guided intraoperative interactive planning. The median V100 (percentage of prostate receiving 100% of the prescribed dose) was 93% (interquartile range (IQR), 91–98%) and the median D90 (minimum dose covering 90% of prostate volume) was 156 Gy (IQR 148–179 Gy).

Patients treated with concomitant ADT or with brachytherapy as a boost were excluded, as well as all Gleason ≥ 4 + 3 or PSA > 20 ng/mL. All patients had to have had a CBC within 3 months before the start of brachytherapy. Patients without a recurrence were included if they had a follow-up of at least 24 months. Only patients with Cancer of the Prostate Risk Assessment (CAPRA) low score (0–2) and intermediate-risk score (3–5) cancers were included. Medication use was recorded at the start of radiotherapy and was not updated following treatment. Because lipid and blood glucose levels, as well as the blood pressure, were not measured at the baseline, we relied on the diagnosis of dyslipidemia, diabetes, or hypertension as a surrogate for patients taking a drug for these conditions.

## 3. Statistical Methods

Biochemical recurrence (BCR) was defined according to the Phoenix definition (nadir prostatic specific antigen (PSA) +2). The log-rank test was used with the Kaplan–Meier method, and Cox regression analysis was employed to assess the univariate association with BCR.

To dichotomize NLR and PLR values, we used previous cut-offs used in the published literature, after verification that these cut-offs were in close proximity to the median values from our cohort. A cut-off level of 150 (median value of this study was 140) for PLR was used [8]. For the same reasons, a cut-off for NLR of 3 was used. In a meta-analysis, Tang et al. [7] found, in a subgroup analysis, that an NLR cut-off value ≥ 3 was a significant prognostic factor in PCa. Furthermore, Keizman et al. [9] utilized an NLR of 3 in their study of metastatic castration-resistant prostate cancer (mCRPC). A review of the literature of many different cancers found that the median cut-off utilized was 4 [6]; however, many others used a value of 3 [10]. The median value in our study was 2.6.

A chi-square test was used to compare distributions between groups. Factors that had a *p* < 0.1 in the univariate analysis were included in multivariable cox regression with stepwise selection to predict the BCR. Statistical significance was defined as *p*-values < 0.05, with Bonferroni corrections used when appropriate. Statistical analysis was performed using SAS Version 9.4.

## 4. Results

### Baseline Characteristics

Patient characteristics of the 604 patients are listed in Table 1. Few patients had abnormal levels of neutrophils (>8 × 10^9^/L *n* = 14, 2.3%), lymphocytes (<1 × 10^9^/L, *n* = 47, 7.8% = 7.6%), or platelets (>450 × 10^9^/L *n* = 6, 1.0%).

The median neutrophil/lymphocyte ratio (NLR) was 6 and the median platelet to lymphocyte ratio (PLR) was 140. The correlation between the NLR and PLR was significant (*r* = 0.726, *p* < 0.001). NLR as a continuous variable was very weakly correlated with age (*r* = 0.08, *p* = 0.056). An NLR > 3 was not more frequent in patients older than 70 years old (46% vs. 39%; *p* = 0.15), or in diabetics; 47% of diabetics had an NLR > 3 compared to 38% of non-diabetics (*p* = 0.079) or in patients taking medication for arterial hypertension (*n* = 289) (42% vs. 38%, *p* = 0.32). However, patients taking a statin were more likely to have an NLR > 3; 45% vs. 36% of patients not taking a statin (*p* = 0.025) (Table 2).

The final results of the multivariate analysis (MVA) for the prediction of biochemical recurrence (BCR). Cancer of the Prostate Risk Assessment (CAPRA) risk group (intermediate vs. low risk, HR 2.15, 95% CI 1.03; 4.47, *p =* 0.0415), as well as NLR > 3 (HR 0.37, 95%CI 0.15; 0.92, *p* = 0.0323), were predictive of BCR. The interaction between the CAPRA risk group and the NLR was not significant (*p* = 0.737) (Table 2).

The median follow-up without recurrence was 60 months (IQR 44–84 months), and the 29 recurrences occurred after a median of 48 months (IQR 29–63). When dividing CAPRA low- and intermediate-risk patients into four subgroups with either an NLR of >3 or ≤3 (Table 3), the only significant difference was between the CAPRA intermediate-risk group with an NLR ≤ 3 (five-year recurrence-free survival of 91%) compared to a 100% recurrence-free survival in patients with CAPRA low-risk and an NLR > 3; (*p* = 0.004, Bonferroni correction for six comparisons *p* = 0.024).

In an alternative exploratory model with only a few patients per analyzed group, using a model including NLR > 3 (*p* = 0.2), PLR > 150 (*p* = 0.3), and statin use (*p* = 0.07), only CAPRA (low- and intermediate-risk) attained statistical significance (*p* = 0.0498).

## 5. Discussion

This study showed that NLR was able to distinguish between more and less aggressive low- and intermediate-risk PCa with significantly different BCR rates. The CAPRA score alone (low- vs. intermediate-risk) was an independent significant predictive factor of BCR. However, in combination with the NLR, we were able to further stratify patients into a group with the worst prognosis: patients with an intermediate risk and an NLR ≤ 3.

Surprisingly, patients with an NLR > 3 had a lower BCR rate. We found that intermediate-risk cancers with an NLR > 3 had similar BCR to patients with low-risk cancers. This is contradictory to the literature, which showed that in many cancers [6], including PCa [7], a higher NLR was associated with a higher rate of recurrence.

Although more than 600 patients were analyzed for the purpose of this study, only 29 biochemical recurrences occurred and this is the main weakness of the present study, decreasing the statistical strength of our results. Recurrences in the analyzed population are generally low. Another weak point is that although the differences between patients with an NLR ≤ 3 vs. > 3 were significant, their clinical impact is small.

To the best of our knowledge, the only previous study to integrate NLR values as a prognostic factor in prostate radiotherapy is that conducted by Langsenlehner et al. [11]. They found that an NLR of ≥5 combined with a Gleason score of <7 vs. ≥7 was an additional prognostic factor. Of their 415 patients, 53.7% were high-risk patients, and thus a significantly different patient population than our cohort. Unfortunately, the median NLR was not available, precluding a direct comparison with values obtained from our study. The study that is probably most comparable to our own included 217 patients who underwent prostatectomy for low-risk prostate cancer and analyzed whether the NLR was predictive of a Gleason score upgrade [12]. With a median NLR of 2.6, the value in this study exactly corresponds to the median value in our study. A key finding was that Gleason ≥ 8 was only found in patients with a preoperative NLR of ≥2.6.

The fact that our results show that a higher NLR is associated with improved outcomes could be explained by the fact that we included only low- and lower-tier intermediate-risk cancers. When looking closely at the meta-analysis by Tang et al. [7], one notes that the data from our previous publication [13] were the least statistically significant. In this previous publication, we included only 11% of patients with a CAPRA score of 6–11, considered as high-risk. The median NLR was 3.0 and higher than in this publication (2.6). A higher NLR seem to be associated with more aggressive cancers and therefore, the relationship between cancer outcomes and blood inflammatory markers may be different in low- to intermediate-risk PCa. 

It is unknown why the NLR exhibits a correlation with different outcome measures in cancer and cardiovascular disease. It is presumed to be a sign of systemic inflammation. However, whether systemic inflammation is associated with local inflammation within the cancer is not clear. There are few studies on the correlation between blood markers of inflammation and local inflammation. Turner et al. [14] found, in colorectal cancers, a non-significant inverse relationship between local and systemic inflammation. In a study of head and neck cancer, PLR and NLR were significantly correlated with the tumor expression of cyclooxygenase-2 (COX2), which is an important enzyme involved in inflammatory reactions [15]. A possible explanation for why an NLR > 3 was a protective factor is that we only selected cancers with a limited aggressiveness in which inflammation and the immune system might act differently. We are aware of only one other study showing that a higher NLR was associated with better overall and melanoma-specific survival in localized stage I-III disease [16]. As in this present study, they found that NLR as a continuous variable was associated with the outcome. In a follow-up study by these authors, the results were not reproducible [17]. Another possible explanation is that local inflammation in the benign prostate can decrease the subsequent incidence of PCa [4,5].

We found that both NLR and PLR were significant predictors of BCR. We focused on NLR because of its widespread use in many cancers and because of the strong correlation between both (0.72, *p* < 0.001). We believe that both have value; the advantage of the NLR is that it can be calculated in a clinical setting much faster than the PLR. Similarly to our study, Huang et al. [18] investigated whether patients with a higher than median white blood cell count treated with chemo-radiotherapy for oropharyngeal cancer would respond differently to treatment. They found that papillomavirus (HPV)-positive patients with a higher than median neutrophil and monocyte count had a worse relapse risk and overall survival in a multivariate analysis (MVA). However, a higher lymphocyte count was associated with a reduced relapse risk. They hypothesized that the negative effect of a higher neutrophil and monocyte count was due to blood-borne bone marrow-derived cells (BMDC), which are recruited in cancers and can counteract the treatment effect. Such an effect has been shown in breast radiotherapy [19] and lung-carcinoma allografts in mice [19,20]. Local irradiation stimulates BMDC infiltration and facilitates recurrence through its effect on tumor vasculature [19].

In a recent publication from our team, including most patients from this study but with a longer follow-up in this present study, we analyzed the influence of the white blood cell count on the overall survival of patients treated with radiation therapy for localized PCa. Because of the limited follow-up, we were not able to find an influence of inflammatory markers on recurrence after radiotherapy. We found that the number of neutrophils, and not the NLR, was an independent prognostic factor for overall survival [13]. In a follow-up study, we found that inflammatory markers are altered by testosterone: High levels of the sum of neutrophils and lymphocytes combined with normal testosterone levels increased the overall mortality [21]. Unfortunately, we only have very few patients with testosterone levels available to study the influence of testosterone on low- and intermediate-risk PCa.

Another interesting finding was that statin users fared better and were more likely to have an NLR > 3. Statin use has been associated with a better BCR rate after radiotherapy and brachytherapy [22,23]. We found that diabetics were not more likely to have an NLR > 3, contradictory to the findings from Kwon et al. [12] based on their patients with clinical low-grade cancers. Although diabetics did not have a lower BCR rate in our study, we previously showed that diabetics taking metformin have a better BCR rate than non-users [24]. Diabetes is associated with chronic systemic inflammation [25]. Both statin and metformin use can have a synergistic effect and reduce BCR after prostatectomy [26].

## 6. Conclusions 

A higher NLR was able to further distinguish between different prognostic groups within patients presenting with low- and intermediate-risk PCa treated with radiotherapy. This finding has to be validated in other cohorts. The effect of the myeloid and lymphoid cell lines in PCa is a promising research field for improving treatment in these patients.

## Figures and Tables

**Table 1 cells-09-02153-t001:** Patient characteristics (*n* = 628).

	Descriptive				Univariate analysis		
	*n*	Frequency (%)	Median	IQR	HR	95%CI	*p*
CAPRA							
Low	355	59			ref		
Intermediate	248	41			2.03	0.96; 4.22	0.054 ^2^
Age			66	61–70	1.02	0.97; 1.08	0.533 ^1^
Neutros			4.2	3.3–5.3	0.85	0.66; 1.08	0.233 ^1^
Lymphos			1.5	1.3–1.9	1.03	0.85; 1.26	0.771 ^1^
Platelets			219	185–256	0.997	0.990; 1.003	0.292 ^1^
HB			147	140–154			
NLR			2.6	2.0–3.7	0.72	0.54; 0.99	**0.046 ^1^**
PLR			140	109–181	0.99	0.99; 1.00	**0.033 ^1^**
NLR >3	241	40			0.39	0.16; 0.97	**0.035 ^2^**
PLR > 150	252	42			0.46	0.21; 1.05	0.058 ^2^
Diabetes	120	20			0.46	0.14; 1.53	0.193 ^2^
Hypertension	276	46			0.84	0.40; 1.76	0.649 ^2^
Statin use	289	48			0.46	0.20; 1.04	0.055 ^2^

^1^ as continuous variable; ^2^ categorical variable. Results in bold are considered statistically significant if *p* < 0.05.

**Table 2 cells-09-02153-t002:** Multivariate analysis predicting biochemical recurrence.

Factor	HR	95% CI	*p*
CAPRA low (ref.) vs. intermediate risk	2.15	1.03; 4.47	0.0415
NLR > 3 (ref.) vs. ≤ 3	0.37	0.15; 0.92	0.0323

**Table 3 cells-09-02153-t003:** CAPRA risk group combined with the neutrophil/lymphocyte ratio (NLR) to predict for BCR (Kaplan–Meier comparison of all groups *p* = 0.0284).

Group	*n*	Number of Recurrences	Percentage Censored	5-Year Recurrence-Free Survival
Low-risk NLR > 3	135	2	98.5	100%
Low-risk NLR ≤ 3	221	11	95.0	97.0%
Intermediate-risk NLR > 3	107	4	96.3	95.0%
Intermediate-risk NLR ≤ 3	141	12	91.5	95.1%

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
