# Peer review of "Systemic Inflammatory Markers Are Predictive of the Response to Brachytherapy in the Prostate"

_cells, 2020, doi:10.3390/cells9102153_

Round 1

Reviewer 1 Report

The paper by Bahig et al entitled “Systemic inflammatory markers can predictive response in radiation therapy to the prostate” reports the influence of the blood inflammatory markers in the complete blood count on recurrence-free survival in low-intermediate risk PCa. In particular, they found that a NLR >3 was more frequent in diabetics and in statin users. By statistical analyses they found that a NLR >3 was associated with a lower rate of BCR while CAPRA low- vs. intermediate risk was not. Indeed, the combination NLR ≤3 score with CAPRA intermediate risk group had the worst BCR rates with a 5-year recurrence-free survival of 91% compared to ≥95% for the other groups.

The manuscript is well designed and presented and the discussion is appropriate reporting other outcomes from literature. I suggest to accept the paper after considering some major revisions:

The Title should be revised as well the abstract. The first sentence in the abstract is not clear the authors should consider to rewrite it.

It is not discussed in the manuscript the potential correlation between NLR and PLR. Is there any data from the literature correlating these two ratio to cancer progression/radiation therpy? Please discuss this point.

Author Response

The Title should be revised as well the abstract. The first sentence in the abstract is not clear the authors should consider to rewrite it.

We changed the title to “Systemic inflammatory markers are predictive of response to brachytherapy in the prostate”

We modified the first sentence of the abstract to “We analyzed the influence of the neutrophil/lymphocyte ratio (NLR) and platelet/lymphocyte ratio (PLR) on biochemical recurrence (BCR) in low-intermediate risk prostate cancer (PCa).”

It is not discussed in the manuscript the potential correlation between NLR and PLR. Is there any data from the literature correlating these two ratio to cancer progression/radiation therpy? Please discuss this point.

We already mentioned in our paper in the results section that “The correlation between the NLR and PLR was significant (r=0.72, p<0.001). We mentioned in the discussion that “We found that both NLR and PLR were significant predictors of BCR. We focused on NLR because of its widespread use in many cancers and because of the strong correlation between both. We believe that both have a value, the advantage of the NLR is that it can be calculated in a clinical setting much faster than the PLR.”

Reviewer 2 Report

This manuscript describes a major finding – that neutrophil/lymphocyte ratio (NLR) determines the propensity for biochemical recurrence (BCR) in intermediate risk patients with prostate cancer undergoing radiotherapy (brachytherapy or EBT). Unlike other cancers where recurrence is observed in patients who have high NLR, in this group, BCR was associated with NLR ≤ 3.0.

  1. The title may need to be altered – from “Systemic inflammatory markers can predictive response in radiation therapy to the prostate” which probably means “Systemic inflammatory markers are predictive of response to radiation therapy in the prostate”
  2. The abstract needs to be reformatted. The period after the first sentence was likely meant to be a comma.
  3. UVA, MVA and CAPRA in the abstract needs to be expanded (e.g. univariate analysis (UVA), multivariate analysis (MVA), etc.)
  4. Table 1 – instead of saying “diabetes”, they need to reevaluate as “Glucose level” or “hemoglobin A1C” or “metformin use” or ‘insulin use” etc. – whichever criterion was used to determine diabetes. Similarly, instead of “hypertension” use actual systolic and diastolic measures or the use of specific drugs.
  5. Add p-value and HR for Table 3 for the different combinations.
  6. While the results, the discussion and the abstract focus on the results of Table 3 in fact, the report of Table 2 which reflect the more traditional outcome (NLR>3 (HR 0.38, 95% CI 0.16; 0.94, p=0.029) indicating prediction of BCR in the NLR > 3 group, has been ignored. It is not clear why.
  7. The explanation for the difference in observation between published literature showing that NLR > 3 is predictive of BCR and this manuscript showing that it is not, needs to be better clarified. They seem to suggest that the difference may lie in blood markers of inflammation vs local inflammation. Evidence of this difference needs to be provided.
  8. In a previous study, the authors used the same patients but with a longer follow up, where they evaluated blood and could do so again. In that study (ref #13) they showed that neutrophil # and not NLR correlated with overall survival. They should probably do so again.

Author Response

  1. The title may need to be altered – from “Systemic inflammatory markers can predictive response in radiation therapy to the prostate” which probably means “Systemic inflammatory markers are predictive of response to radiation therapy in the prostate”Thank you for this suggestion, we used it for our title
  2.  
  3.  
  4. The abstract needs to be reformatted. The period after the first sentence was likely meant to be a comma. 
  5. We hope that this first sentence is now much clearer: “We analyzed the influence of the neutrophil/lymphocyte ratio (NLR) and platelet/lymphocyte ratio (PLR) on biochemical recurrence (BCR) in low-intermediate risk prostate cancer (PCa).”
  6. UVA, MVA and CAPRA in the abstract needs to be expanded (e.g. univariate analysis (UVA), multivariate analysis (MVA), etc.)
  7. We explained these abbreviations in the abstract.
  8. Table 1 – instead of saying “diabetes”, they need to reevaluate as “Glucose level” or “hemoglobin A1C” or “metformin use” or ‘insulin use” etc. – whichever criterion was used to determine diabetes. Similarly, instead of “hypertension” use actual systolic and diastolic measures or the use of specific drugs.
  9. We relied on pre-treatment diagnosis by their treating physicians for the diagnosis of diabetes, hypertension and dyslipidemia. We mentioned this in the methods section.
  10. Add p-value and HR for Table 3 for the different combinations.
  11.  
  12. While the results, the discussion and the abstract focus on the results of Table 3 in fact, the report of Table 2 which reflect the more traditional outcome (NLR>3 (HR 0.38, 95% CI 0.16; 0.94, p=0.029) indicating prediction of BCR in the NLR > 3 group, has been ignored. It is not clear why. 
  13. We wanted to show that the NLR helps to refine the predictive value of the traditional risk groups such as the CAPRA.. Therefore we didn’t put much weight on Table 2, more on table 3. We noted at the beginning of the discussion section that “... in combination with the NLR, we were able to further stratify patients into a group with worse prognosis than low-risk cancers: intermediate risk and an NLR <3.”
  14. The explanation for the difference in observation between published literature showing that NLR > 3 is predictive of BCR and this manuscript showing that it is not, needs to be better clarified. They seem to suggest that the difference may lie in blood markers of inflammation vs local inflammation. Evidence of this difference needs to be provided.
  15. In fact our study showed that NLR was predictive of biochemical recurrence, only contrary to what was expected: a higher NLR was a positive predictive factor. There is no evidence proving a link between markers of systemic inflammation such as the NLR and local inflammation. We mentioned this before in our manuscript saying that “There are few studies about the correlation between blood markers of inflammation and local inflammation. Turner et al. (14) found in colorectal cancers a non-significant inverse relationship between local and systemic inflammation
  16. In a previous study, the authors used the same patients but with a longer follow up, where they evaluated blood and could do so again. In that study (ref #13) they showed that neutrophil # and not NLR correlated with overall survival. They should probably do so again. 
  17. In fact we did, table 1 listed neutrophils, leucocytes as well as platelet count in univariate analysis. None of these factors was predictive of biochemical recurrence.

Reviewer 3 Report

The treatment method and total dose of the radiation therapy are very important in the radical cure treatment to prostate cancer. This article does not mention it and mixes various kinds of conditions about radiotherapy.

If the systemic inflammatory markers can predict the prognosis, it is necessary to unify background factors about the radiotherapy modality, and to report it. The external beam irradiation and brachytherapy should be distinguished. 

It is necessary for the author to examine external beam irradiation and brachytherapy separately.

The treatment response and prognosis of the PCa depend on the radiation therapy modality and the irradiated dose. There is not mention about the dose of the radiation therapy at all.

I cannot recommend publication of this manuscript.

Author Response

The treatment method and total dose of the radiation therapy are very important in the radical cure treatment to prostate cancer. This article does not mention it and mixes various kinds of conditions about radiotherapy.

We eliminated the 23 patients treated with EBRT from our analysis. CAPRA score became now a significant factor in multivariate analysis. We don’t think that our results changed significantly except for the CAPRA score. We hope that this reviewer sees our paper now more favorably

If the systemic inflammatory markers can predict the prognosis, it is necessary to unify background factors about the radiotherapy modality, and to report it. The external beam irradiation and brachytherapy should be distinguished. 

It is necessary for the author to examine external beam irradiation and brachytherapy separately.

Please see our answer above, we eliminated the EBRT patients.

The treatment response and prognosis of the PCa depend on the radiation therapy modality and the irradiated dose. There is not mention about the dose of the radiation therapy at all.

Please see our answer above, we eliminated the EBRT patients.

I cannot recommend publication of this manuscript.

We hope that this reviewer sees our paper now more favorably

Round 2

Reviewer 1 Report

The authors have now improved the manuscript satisfying the requests. 

Author Response

NA

Reviewer 2 Report

The authors have responded to most of the questions. 

Author Response

NA

Reviewer 3 Report

 This paper is written by radiation oncologist, but the details of brachytherapy protocol are absent. This is mysterious. The details of brachytherapy are useful for the readers of radiation oncologist. Furthermore this study may be useful for urologist.

Minor comment

Abstract;Methods “604 patients treated with,,,,”

 The number of the beginning of a sentence becomes the spelling notation with any number.

Author Response

This paper is written by radiation oncologist, but the details of brachytherapy protocol are absent. This is mysterious. The details of brachytherapy are useful for the readers of radiation oncologist. Furthermore this study may be useful for urologist.

There is nothing sinister with us not mentioning details of prostate brachytherapy. We thought that it was not pertinent to the subject.

We added in the methods section that “Two physicians performed all brachytherapy implants with Iodine-125 seeds via 3-D trans-rectal ultrasound-guided intraoperative interactive planning. The median V100 (percentage of prostate receiving 100% of the prescribed dose) was 93% (interquartile range, IQR, 91-98%) and the median D90 (minimum dose covering 90% of prostate volume) was 156 Gy (IQR 148-179 Gy)”.

Minor comment

Abstract;Methods “604 patients treated with,,,,”

 The number of the beginning of a sentence becomes the spelling notation with any number.

Thank you for pointing this out. We changed the sentence to “We treated 604 patients with ….”